# Benchmarking and Enhancing LLM Agents in Localizing Linux Kernel Bugs

## Abstract

The Linux kernel is a critical system, serving as the foundation for numerous systems. Bugs in the Linux kernel can cause serious consequences, affecting billions of users. Fault localization (FL), which aims at identifying the buggy code elements in software, plays an essential role in software quality assurance. While recent LLM agents have achieved promising accuracy in FL on recent benchmarks like SWE-bench, it remains unclear how well these methods perform in the Linux kernel, where FL is much more challenging due to the large-scale code base, limited observability, and diverse impact factors. In this paper, we introduce LinuxFLBench, a FL benchmark constructed from real-world Linux kernel bugs. We conduct an empirical study to assess the performance of state-of-the-art LLM agents on the Linux kernel. Our initial results reveal that existing agents struggle with this task, achieving a best top-1 accuracy of only 41.6% at file level. To address this challenge, we propose LinuxFL$^+$, an enhancement framework designed to improve FL effectiveness of LLM agents for the Linux kernel. LinuxFL$^+$ substantially improves the FL accuracy of all studied agents (e.g., 7.2% - 11.2% accuracy increase) with minimal costs.

## 1 Introduction

The Linux kernel is a critical system which serves as the foundation for numerous operating systems, servers, and embedded systems, and has evolved over decades with contributions from thousands of developers and billions of users (The Linux Foundation, 2020). Given the widespread adoption of the Linux kernel, bugs in the Linux kernel can cause serious consequences, affecting a vast number of users. Therefore, extensive research has been dedicated to developing automated software quality assurance techniques (e.g., testing (Bligh & Whitcroft, 2006; Chen et al., 2013; Yang et al., 2025b) and debugging (Bissyandé et al., 2012; Edge, 2019; Serrano et al., 2020; Jeong et al., 2023)) specifically for the Linux kernel.

Fault localization (FL), which aims at identifying the buggy code elements (e.g., files or functions) in software, plays a critical role in software quality assurance. Given the codebase of the buggy software and the bug report (e.g., a user-reported bug symptom description), automated FL techniques return a list of buggy code elements ranked by their suspiciousness (i.e., the probability of being buggy). In particular, accurate FL is a prerequisite for bug fixing, as a bug cannot be resolved without correctly identifying the faulty code location.

Traditional FL techniques mainly leverage heuristics (Abreu et al., 2006; Wong et al., 2014b) or information retrieval (IR) (Zhou et al., 2012; Saha et al., 2013) to identify buggy code elements. More recently, with the advance in large language models (LLMs), LLM agents Liu et al. (2024) have demonstrated remarkable accuracy in FL. Equipped with tool invocation, agents can autonomously navigate codebases to identify the buggy location. For example, the state-of-the-art agents such as SWE-Agent (Yang et al., 2024), AutoCodeRover (Zhang et al., 2024), Agentless (Xia et al., 2024), achieve around 70% accuracy in localizing buggy files for Python software in the benchmark SWE-bench (Jimenez et al., 2024).

Although achieving promising FL effectiveness, existing agents have been mainly evaluated on general software at moderate scales. It remains unclear how existing agents perform in complex, large-scale software systems like the Linux kernel. In particular, FL in Linux

kernel is more challenging than general software due to the following factors. (1) *Large-scale Codebase*: the Linux kernel has a massive codebase significantly larger than general software. For example, the v5.8 release of Linux kernel includes over 69K files and 28M lines of code (The Linux Foundation, 2020), which is over 30 times the scale of even the largest project in the most widely-used benchmark SWE-bench. (2) *Limited Observability*: given the real-time nature of the Linux kernel with the need to minimize overhead, the kernel restricts the use of instrumentation and logging mechanisms during runtime. Additionally, the kernel operates in a privileged mode, isolated from user space. As a result, user-reported bug descriptions often lack detailed runtime information and debugging hints, creating a significant gap between the user description and the actual root causes. (3) *Diverse Impact Factors:* kernel bugs are influenced by a wide range of factors, including hardware variability (e.g., architectural configurations) and runtime variability (e.g., system load or timing). These factors lead to an exponentially large reasoning space to accurately diagnose the root causes of errors. Given the unique challenges and the importance of the kernel, this work aims at investigating the FL effectiveness of state-of-the-art LLM agents on the Linux kernel.

**Benchmark.** We first build a new benchmark LINUXFLBENCH of 250 real-world FL tasks for the Linux kernel. Each FL task in LINUXFLBENCH includes a user-submitted bug report, the buggy Linux kernel codebase, and the ground-truth buggy locations based on the associated commit patches. LINUXFLBENCH involves a wide range of Linux kernel bugs, spanning over 120 Linux kernel versions and 66 different kernel components. The FL tasks are significantly more challenging than those in SWE-bench, as evidenced by the substantially larger codebases (10–30× more files and lines of code) and more complex bug reports (approximately 1.5× more words).

**Empirical Study.** On LINUXFLBENCH, we make the first attempt to evaluate state-of-the-art LLM agents in localizing Linux kernel bugs. Our results reveal the limited FL effectiveness (e.g., 36.8% - 41.6% accuracy) of existing agents in the Linux kernel; such a FL accuracy is much lower than their performance on general software systems (a 16.7% - 31.9% accuracy drop from SWE-bench). We further perform bad case analysis and find that existing agents mainly miss the buggy files as they fail to capture the related files or to cover complete root causes of kernel bugs. The results indicate that FL in the Linux kernel is indeed a more challenging task, highlighting the need for building more advanced agents to localize bugs in large and complex software systems like the Linux kernel.

**Technique.** Inspired by our study above, we further propose an enhancing framework LINUXFL$^+$, which improves the FL effectiveness of existing agents for the Linux kernel. LINUXFL$^+$ incorporates two expansion strategies to refine the prediction results of existing agents: directory-aware expansion to include buggy files based on the repository structure, and potential cause expansion to identify buggy files based on the additional bug knowledge from Linux kernel mailing list (LKML) (Kernel.org, 2025b). Our evaluation results show that LINUXFL$^+$ can substantially improve the FL accuracy of all studied agents (e.g., 7.2% - 11.2% accuracy increase) with minimal costs. Moreover, the ablation analysis confirms the contribution of each expansion strategies.

## 2 BACKGROUND AND RELATED WORK

**FL Task Definition.** Given the bug report and codebase, FL techniques identify buggy code elements (e.g., files or functions). Formally, let a codebase be represented as a set of code elements, $\mathcal{C} = \{ce_1, ce_2, \ldots, ce_N\}$, where $N$ denotes the total number of code elements. A bug report $BR$ typically includes a title, a description, and optional metadata (e.g., component and hardware information in the context of Linux kernel), and can be expressed as $BR = (title, desc, meta)$. A FL task can be modeled as: **FL** $: BR, \mathcal{C} \rightarrow list(\mathcal{C})$, where $list(\mathcal{C})$ denotes a list of code elements that ranked by their probabilities of being buggy.

**Existing FL techniques.** FL techniques have been extensively studied in literature:

- **Coverage-based FL.** Besides bug reports, some FL techniques leverage test coverage to identify buggy locations, such as SBFL (Abreu et al., 2006; Wong et al., 2014b), GNN-based FL (Lou et al., 2021), AutoFL (Kang et al., 2024), and AgentFL (Qin et al., 2024).

Table 1: Existing Benchmarks for Software Maintenance

| Benchmark | Language | # Repo | # Bugs | Data Source | Linux-Related | User-reported |
|---|---|---|---|---|---|---|
| Defects4J (Just et al., 2014) | Java | 17 | 854 | Bug Tracking Systems | ✗ | ✓ |
| Linux-3.16 (Saha et al., 2014) | C | 1 | 1,548 | Bug Tracking Systems | ✓ | ✓ |
| SWE-bench (Jimenez et al., 2024) | Python | 12 | 2,294 | GitHub Pull Requests | ✗ | ✓ |
| FAUN-Eval-fix (Hu et al., 2024) | Multiple | 17 | 300 | GitHub Pull Requests/Issues | ✗ | ✓ |
| KBENCHSYZ (Mathai et al., 2024) | C | 113 | 279 | Fuzzing-Detected Crashes | ✓ | ✗ |
| Loc-Bench (Chen et al., 2025) | Python | 165 | 560 | GitHub Issues | ✗ | ✓ |
| SWE-lancer (Miserendino et al., 2025) | Python | 1 | 1,488 | Upwork Issues | ✗ | ✓ |
| **LinuxFLBench** | C | 120 | 250 | Bug Tracking Systems | ✓ | ✓ |

However, coverage and executable failure-triggering tests are not always available in practice. Especially for the large systems like Linux kernel, users report bugs by textually describing the error symptoms. Therefore, coverage-based FL cannot be applied to the Linux kernel when only bug reports are available, which thus is not included in this work.

- **Information Retrieval (IR) Based FL.** FL can be formulated as an information retrieval (IR) problem, where a bug report serves as a query to rank code files by relevance. Existing IR-based FL techniques use various similarity measures, such as Vector Space Model (VSM) (Zhou et al., 2012; Saha et al., 2013; 2014; Wang & Lo, 2014; Wong et al., 2014a), Dirichlet Language Model (DLM) (Sisman et al., 2017), or deep learning approaches (Huo et al., 2021; Ciborowska & Damevski, 2022; Mohsen et al., 2023). In this work, we empirically evaluate IR-based FL in the Linux kernel.
- **Agent-based FL.** Recent advances in LLM agents have shown strong performance in software maintenance tasks, including FL. For instance, SWE-Agent (Yang et al., 2024) incorporates a custom-built Agent-Computer Interface to navigate entire repositories; AutoCodeRover (Zhang et al., 2024) equips LLMs with code search capabilities to retrieve relevant code contexts; Agentless (Xia et al., 2024) refines the localization process by restricting the decision-making autonomy of agents. In this work, we not only make the first attempt to empirically evaluate existing agents in the Linux kernel, but also propose a framework to enhance their performance in this challenging domain.

**Benchmarks for Software Maintenance.** As FL is a key sub-task in software maintenance, we revisit existing software maintenance benchmarks in Table 1. The majority of existing benchmarks focus on general software systems in Java or Python. In contrast, our benchmark LinuxFLBench specifically targets the large-scale system Linux kernel. Only two prior benchmarks involve the kernel: Linux-3.16 (Saha et al., 2014), which is limited to a single old version, and KBENCHSYZ (Mathai et al., 2024), which collects Syzkaller (Google, 2025)-detected crash bugs. LinuxFLBench differs by (1) covering a wider range of kernel versions, (2) including diverse real-world bug types beyond crashes (e.g., functionality and performance bugs), and (3) sourcing all bugs from user reports rather than automated fuzzing. Thus, LinuxFLBench complements existing efforts by offering a more comprehensive benchmark for evaluating advanced FL techniques in the Linux kernel.

## 3 LinuxFLBench: A FL Benchmark for Linux Kernel

LinuxFLBench is a new benchmark of 250 real-world Linux kernel FL tasks.

### 3.1 Construction of LinuxFLBench

LinuxFLBench is constructed through three phases, as described in Appendix B.1.

**Step 1: Bug Report Collection.** We collected Linux kernel bug reports from Kernel.org Bugzilla (Kernel.org, 2025a) up to December 31, 2024. Each report includes a *title*, *description*, and relevant *metadata* (e.g., kernel version, environment). To ensure code availability, we retained only reports linked to kernel versions hosted on the official Linux website (Kernel.org, 2025c). For ground-truth reliability, we required reports marked as "CLOSED" and "CODE_FIX" in the bug tracking system. Furthermore, we included only bug reports with patches attached, enabling us to identify the buggy locations based on the patch information. In total, we collected 2,138 bug reports during this step.

**Step 2: Buggy Location Identification.** For each collected bug report, we identified the location modified in the developer-committed patch as the ground-truth buggy location. Specifically, we traversed source files with the extensions `.c` or `.h`, skipping other file types

such as `README` or `Makefile`. Following SWE-bench-lite (Jimenez et al., 2024), we kept only unambiguous cases where exactly one file was modified to ensured the reliability of the ground truth. After this step, 635 bug reports with identified buggy files were obtained.

**Step 3: Manual Inspection.** To further ensure quality, we manually reviewed the collected data. Three human annotators checked each bug as follows: (1) bug reports without actual bugs (e.g., those that primarily submit patches) were excluded; (2) bug reports with sufficient information (e.g., clear natural language descriptions or detailed system logs) were retained; (3) bug reports that explicitly mentioned buggy locations or fix solutions were excluded. As a result, the final dataset comprises 250 high-quality FL tasks, and each task includes a bug report, the buggy codebase, and the ground-truth buggy file and method(s). A detailed sample is shown in Appendix B.2.

## 3.2 CHARACTERISTICS OF LINUXFLBENCH

LINUXFLBENCH presents challenging tasks with complex bug reports and large-scale codebase, offering multidimensional diversity across kernel versions, products, and bug types.

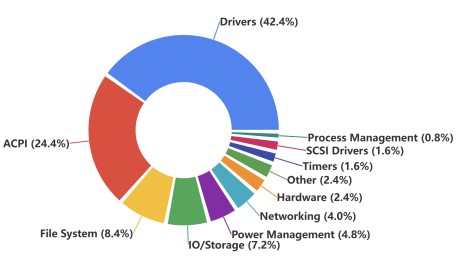

Figure 1: Task Distribution across Products

Table 2: Task Scales of LINUXFLBENCH and SWE-bench.

| | Benchmark | Bug Description | Codebase | |
|---|---|---|---|---|
| | | # Words | # Files | # Lines |
| Mean | LINUXFLBENCH | 283.1 | 28,808 | 11,492K |
| | SWE-bench* | 195.1 | 3,010 | 438K |
| Max | LINUXFLBENCH | 5,139 | 67,073 | 28,178K |
| | SWE-bench* | 4,477 | 5,890 | 886K |

* Source: SWE-bench (Jimenez et al., 2024).

**Scale.** Table 2 compares the scale of tasks in LINUXFLBENCH and SWE-bench. Our dataset is more challenging, with the codebase tens of times larger and bug reports that are more detailed and complex. We also compare stack trace lengths, the presence of bug location in bug reports, and the sizes of buggy files and golden patches, all of which further underscore the greater complexity of our dataset. More details are provided in Appendix B.3.

**Products.** Fig. 1 shows the distribution of LINUXFLBENCH across kernel products (i.e., high-level categories defined in Bugzilla). In particular, bugs span 16 products, with *Drivers*, *ACPI*, and *File System* being the largest categories. At a finer granularity, the benchmark covers a diverse set of 66 components, with the most frequent being *network-wireless* (6.4%), *Video* (6.0%), *Network* (5.2%), *Power-Battery* (4.8%), and *Sound* (4.4%).

**Versions.** The Linux kernel has evolved over several decades, resulting in the release of numerous versions. LINUXFLBENCH captures this temporal diversity by including bugs from a broad range of kernel versions, covering a total of 120 distinct versions.

**Bug Types.** LINUXFLBENCH encompasses a broad spectrum of bugs by symptoms and causes. Symptomatically, it includes common issues such as system crashes (14.8%), power malfunctions (13.6%), and network failures (10.8%). Causally, frequent sources are hardware configuration faults (19.6%), memory defects (15.6%), and data handling errors (15.2%).

## 4 EVALUATION OF LLM AGENTS ON LINUXFLBENCH

We empirically evaluate SOTA LLM agents on LINUXFLBENCH to investigate their FL effectiveness in the Linux kernel.

### 4.1 STUDY SETUP

**Studied Baselines.** (1) *LLM agents.* We study three SOTA LLM agents, i.e., SWE-Agent (Yang et al., 2024), AutoCodeRover (Zhang et al., 2024), and Agentless (Xia et al.,

2024), as they are fully open-sourced and achieve high effectiveness in recent software maintenance leaderboard (SWE-bench, 2025). All agents are equipped with GPT-4o (gpt-4o-2024-08-06) as backbone LLMs (OpenAI, 2024). The detailed implementation of these agents is in Appendix C. (2) *IR-based baselines.* For comparison, we also include traditional IR-based FL baselines for comparison. Specifically, we include the classic IR-based methods BugLocator (Zhou et al., 2012) and BLUiR (Saha et al., 2013), along with widely used IR techniques such as BM25 (Robertson et al., 1995) and Sentence-BERT (Reimers & Gurevych, 2019).

**Evaluation Metrics.** In line with previous FL work (Xia & Lo, 2023; Zhou et al., 2012; Saha et al., 2014), we include the widely-used metrics like recall at top-k (k = 1, 5, 10) and the Mean Reciprocal Rank (MRR) to evaluate the FL effectiveness.

## 4.2 QUANTITATIVE ANALYSIS

Table 3 shows the overall file-level FL effectiveness of studied techniques on LINUXFLBENCH.

**Comparison with IR-based methods.** Overall, existing agents outperform all traditional IR methods, indicating the benefits from agentic solutions in identifying buggy locations for large scale systems. For instance, SWE-Agent achieves the best effectiveness with an MRR of 0.476, significantly surpassing other methods. Among IR methods, BLUiR performs the best, but only with an MRR of 0.321.

Table 3: FL effectiveness on LINUXFLBENCH.

| Methods | Recall@1 | Recall@5 | Recall@10 | MRR |
|---|---|---|---|---|
| BM25 | 0.168 | 0.328 | 0.396 | 0.231 |
| BugLocator | 0.127 | 0.209 | 0.272 | 0.215 |
| BLUiR | 0.228 | 0.317 | 0.404 | 0.321 |
| Sentence-BERT | 0.056 | 0.136 | 0.180 | 0.090 |
| SWE-Agent | **0.416** | **0.552** | **0.584** | **0.476** |
| AutoCodeRover | 0.388 | 0.496 | 0.496 | 0.435 |
| Agentless | 0.368 | 0.492 | 0.504 | 0.419 |

**Comparison with general software system.** Although outperforming traditional IR methods, existing agents still exhibit limited overall effectiveness on Linux kernel. For instance, even the best-performing SWE-Agent only achieves a top-1 recall of only 0.416 on LINUXFLBENCH, which is much lower than when it is applied to general software systems (i.e., SWE-bench). In particular, Fig.2 compares the FL effectiveness of agents in Linux systems (i.e., on LINUXFLBENCH) and in general software systems (i.e., on SWE-bench). The reported SWE-bench results are from previous work (Xia et al., 2024). We can observe a marked performance decline for all the LLM agents on LINUXFLBENCH compared to SWE-bench, with recall values decreasing by more than 0.15. Such an effectiveness drop underscores the heightened challenges associated with FL in the larger and more intricate Linux kernel codebase than general software systems.

**Uniqueness and Union.** Fig. 3 presents the overlapped/unique bugs that are correctly localized at top-1 by studied agents. We could observe complementary strengths of the different approaches, as each agent can uniquely resolve 12 - 20 bugs. Nevertheless, even when combining the correctly-localized bugs of all agents, only 146 bugs out of 250 total bugs can be successfully localized (i.e., 58.4% top-1 recall). It further highlights the considerable challenges that agents still face in performing FL within the complex Linux kernel.

## 4.3 QUALITATIVE ANALYSIS

To further understand why agents perform poorly in Linux kernel, we manually examine bad cases where all studied agents fail to correctly localize the buggy files. Overall, we find two main reasons for the limited effectiveness as follows.

**Confusion Among Related Files.** As a large-scale software system, bugs in Linux kernel often propagate along a long chain, where many related files are associated with each other via function calls or data dependencies. While agents might be capable of coarse-grained FL (e.g., correctly identifying the buggy directories or high-level modules), they struggle to further precisely pinpoint the exact faulty file/method among all the related files. This challenge is indirectly evidenced by the fact that each Linux directory in LINUXFLBENCH contains, on average, approximately twice as many files (16 vs. 8) as those in SWE-bench, making fine-grained localization within directories more difficult. For example, Appendix D.1 shows a bad case where all agents wrongly localize the files that are in the same directory as the buggy file.

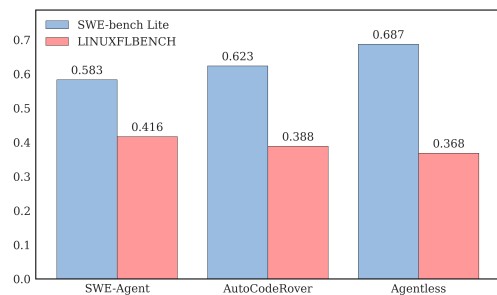

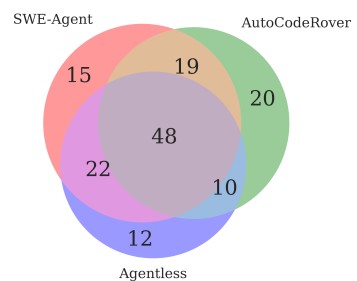

Figure 2: Performance of LLM agents on SWE-bench and LinuxFLBench.

Figure 3: Venn Diagram for Correctly Localized Bugs by LLM agents.

**Limited Exploration of Potential Causes.** Given the complexity of the Linux kernel, a bug can arise from diverse and non-obvious root causes. Current agents narrowly focus on a small set of highly probable causes, failing to explore a broader range of potential causes. Consequently, this limited exploration leads to missed opportunities for correctly identifying the buggy file. Appendix D.2 shows a bad case that all agents miss the real cause.

## 5 LinuxFL+: An Enhancing Framework

To address the limitations of existing agent-based methods, we propose a novel enhancing framework LinuxFL+, which improves the FL effectiveness of agents in the Linux kernel.

### 5.1 Approach

As discussed in Section 4.3, given the huge space of Linux kernel, existing agents fail to capture the relationship between files or to cover a complete pool of potential causes. Therefore, the main insight of LinuxFL+ is to *expand* the prediction results of existing agents with both the repository structure and the root causes.

Fig. 4 shows the overall workflow of LinuxFL+. Given the buggy files predicted by any agent (e.g., AutoCodeRover), LinuxFL+ refines the prediction via the following three phases. (1) *Directory-Aware Expansion*: LinuxFL+ expands the search scope within directories of the initial predictions generated by LLM agents. LinuxFL+ then re-selects bug-related files within these directories, enabling a more thorough exploration of related files; (2) *Potential Cause Expansion*: LinuxFL+ explores as many potential causes as possible to scale the related files. LinuxFL+ includes two hypothesizing strategies to expand the potential causes for the given bug report, leveraging both the original capabilities of LLMs (i.e., direct hypothesis) and the additional knowledge from Linux kernel mailing list (i.e., mail-augmented hypothesis); (3) *Candidate Integration*: all relevant files are merged as candidates, followed by a re-ranking process to further refine the results.

#### 5.1.1 Directory-Aware Expansion

While existing agents can generally identify the correct modules related to a bug, they often struggle to distinguish relevant files within those modules. To address this limitation, LinuxFL+ first expands the search scope to include all files in the directories of the initially predicted files. Using this expanded candidate set, the LLM re-selects files likely related to the bug. We retain the top-k (k=10) most relevant files as the expanded results. This approach provides the LLM with an additional opportunity to identify buggy files, enabling a more comprehensive exploration of related files. Detailed prompts are in Appendix E.1.

#### 5.1.2 Potential Cause Expansion

Current agents tend to focus narrowly on few highly probable causes within limited steps. However, diagnosing complex bugs often requires an iterative "guess-and-check" process (Alaboudi & LaToza, 2023; Layman et al., 2013; Liu et al., 2025), where developers form

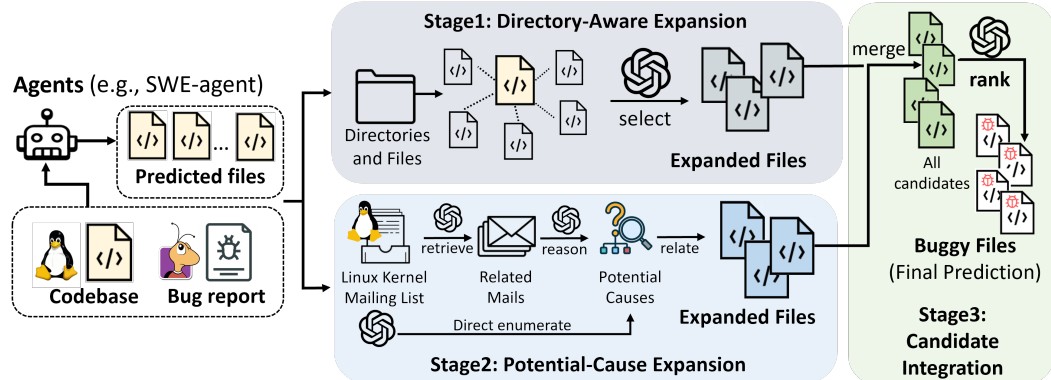

Figure 4: Overview of LINUXFL+.

experience-based hypotheses and progressively refine their understanding to isolate the root cause. Inspired by this, we expand bug-related files by exploring a broader range of potential causes. Specifically, we design two types of hypothesizing strategies to expand probable causes: *Direct Hypothesis*, leveraging models' inherent knowledge on Linux kernel, and *Mail-Augmented Hypothesis*, integrating historical bug knowledge from mailing list discussions.

**Direct Hypothesis.** As LLMs already possess a foundational understanding of the Linux kernel from extensive pre-training, a straightforward expansion approach is to fully leverage the intrinsic knowledge of models. To this end, we design prompts that instruct the model to generate plausible potential causes, and rank these causes based on their estimated likelihood of being responsible for the bug. To ensure the practicality of each hypothesized cause, the LLM is also required to propose a corresponding fix and identify the specific files that would need modification. We then extract the predicted target files, preserving their original ranking from the associated causes. Detailed prompts are in Appendix E.2.

**Mail-Augmented Hypothesis.** Relying solely on the intrinsic capabilities of LLMs is insufficient, as general-purpose models still lack in-depth and domain-specific knowledge of Linux kernel. To address this limitation, we incorporate historical bug knowledge from the Linux kernel mailing list (LKML) (Kernel.org, 2025b). The LKML is the communication channel among Linux kernel developers, including massive emails discussing bugs, patches, and diverse topics on maintaining Linux kernel. Specifically, we adopt a Retrieval-Augmented Generation (RAG) approach, using mailing list data as an external knowledge base to provide more comprehensive and diverse bug causes in Linux kernel.

*Mail Collection.* To construct the kernel knowledge base, we first collect emails from the LKML. We retain only emails that include patches, as these are more likely to involve discussions of bug fixes or feature implementations, providing useful context for FL. To ensure quality, we discard non-atomic patches modifying over 10 files, as these typically represent merged changes. Additionally, to avoid potential data leakage, we exclude any emails containing external URLs or the keyword "bugzilla".

*Mail Retrieval.* We adopt a hierarchical retrieval strategy: (1) restrict the search space to only emails linked to code files predicted by agents, and (2) reformulate noisy bug reports(e.g., hexadecimal logs) into four key dimensions—bug behavior, potential causes, expected behavior, and possible solutions. We then apply BM25 (Lù, 2024) to retrieve the top-10 relevant emails restricted to those sent before the bug report for temporal consistency.

*Mail-augmented hypothesis.* Using retrieved mails, we prompt LLMs to generate more diverse and informed causes for the bug, which in turn guide the identification of related buggy files. This step is similar to *Direct Hypothesis* but augmented with mail knowledge. Detailed prompts are in Appendix E.2.

### 5.1.3 CANDIDATE INTEGRATION

In this final phase, we consolidate the files predicted by previous two expansion strategies and rank the aggregated candidate files to produce the final FL results.

We adopt a simple yet effective merging strategy. Specifically, for each candidate file $f$, we collect its ranks from the three sources: $R_{dir}(f)$ (Directory-Aware Expansion), $R_{direct}(f)$ (Direct Hypothesis), and $R_{mail}(f)$ (Mail-Augmented Hypothesis). If a file does not appear in the results of a particular method, its rank is set to $\infty$. We then compute an aggregated score for $f$ as follows: $\text{score}(f) = \frac{1}{R_{dir}(f)} + \frac{1}{R_{direct}(f)} + \frac{1}{R_{mail}(f)}$. Files that achieve better ranks in any individual method receive higher scores, while those consistently ranked highly across methods are further prioritized. All candidate files are sorted by their aggregated scores to produce the initial merged ranking. To further refine this list, the LLM is prompted to re-rank the files based on the semantic correspondence between their path and bug report.

## 5.2 Experimental Setup

**Baselines.** To evaluate the effectiveness of LinuxFL$^+$ in improving existing agents, we apply LinuxFL$^+$ to refine the prediction outputs of recent agents (i.e., SWE-Agent, AutoCodeRover, and Agentless) on LinuxFLBench.

**Implementation Details.** We leverage GPT-4o (OpenAI, 2024) (gpt-4o-2024-08-06) and the open-source Qwen3-32B (Yang et al., 2025a) as the backbone models for implementing LinuxFL$^+$. We configure the model temperature as 0 to ensure relatively deterministic outputs with other parameters as default settings.

## 5.3 Results and Analysis

### 5.3.1 Overall Performance

Table 4: Evaluation results of LinuxFL$^+$.

| Methods | Recall@1 | Recall@5 | Recall@10 | MRR |
|---|---|---|---|---|
| SWE-Agent | 0.416 | 0.552 | 0.584 | 0.476 |
| - w/ LinuxFL$^+$ (GPT-4o) | 0.524 (+0.108) | 0.720 (+0.168) | 0.768 (+0.184) | 0.610 (+0.134) |
| - w/ LinuxFL$^+$ (Qwen3-32B) | 0.476 (+0.060) | 0.664 (+0.112) | 0.704 (+0.120) | 0.558 (+0.082) |
| AutoCodeRover | 0.388 | 0.496 | 0.496 | 0.435 |
| - w/ LinuxFL$^+$ (GPT-4o) | 0.500 (+0.112) | 0.712 (+0.216) | 0.744 (+0.248) | 0.589 (+0.154) |
| - w/ LinuxFL$^+$ (Qwen3-32B) | 0.440 (+0.052) | 0.664 (+0.168) | 0.720 (+0.224) | 0.539 (+0.105) |
| Agentless | 0.368 | 0.492 | 0.504 | 0.419 |
| - w/ LinuxFL$^+$ (GPT-4o) | 0.440 (+0.072) | 0.684 (+0.192) | 0.724 (+0.220) | 0.549 (+0.130) |
| - w/ LinuxFL$^+$ (Qwen3-32B) | 0.432 (+0.064) | 0.652 (+0.160) | 0.688 (+0.184) | 0.525 (+0.106) |

Table 5: Cost of LinuxFL$^+$.

| Methods | # Tokens | $ Cost |
|---|---|---|
| SWE-Agent | 72.4 K | 0.194 |
| - w/ LinuxFL$^+$ | 14.0 K | 0.041 |
| AutoCodeRover | 206.6 K | 0.560 |
| - w/ LinuxFL$^+$ | 11.8 K | 0.035 |
| Agentless | 150.2 K | 0.396 |
| - w/ LinuxFL$^+$ | 15.3 K | 0.044 |

Table 4 presents the improvements of LinuxFL$^+$ on all studied agents.

**Effectiveness.** LinuxFL$^+$ exhibits strong performance in enhancing the FL capabilities of agents, as evidenced by substantial improvement across all evaluation metrics. For example, when applied to SWE-Agent with GPT-4o, Recall@10 increases from 0.584 to 0.768, an absolute gain of 18.4 percentage points. Moreover, Recall@1 improves by 10.8 percentage points (from 0.416 to 0.524). The improvement indicates the effectiveness of the expansion strategies of LinuxFL$^+$, which successfully recover the buggy files missed by existing agents.

**Generalizability.** LinuxFL$^+$ consistently enhances performance across all state-of-the-art agents and remains effective with different LLMs. Notably, agents with relatively weaker baselines, such as AutoCodeRover and Agentless, achieve performance comparable to SWE-Agent once integrated with LinuxFL$^+$. Furthermore, LinuxFL$^+$ yields consistent gains even when applied to smaller open-source models such as Qwen3-32B. These results highlight the strong generalizability of LinuxFL$^+$ and its effectiveness across agent-based approaches with diverse baseline strengths and LLM capacities.

**Ablation study.** We perform an ablation study to investigate the contribution of each component in LinuxFL$^+$. In particular, we find all the expansion strategies, i.e., directory-aware expansion and potential causes expansion (with either direct or mail-augmented hypothesis) can improve the FL effectiveness of agents. Detailed results can be found in Appendix F.

**Cost-efficiency.** Table 5 presents the cost of applying LinuxFL$^+$ on LinuxFLBench with GPT-4o. As shown, while LinuxFL$^+$ achieves strong performance, it incurs only a modest additional cost. On average, the total number of tokens used per task by LinuxFL$^+$ ranges from 11.8k to 15.3k, resulting in an estimated cost of approximately \$0.04. This is roughly one-tenth of the cost incurred by agent-based baselines. The primary cost of LinuxFL$^+$

stems from its use of email content. These results suggest that LINUXFL⁺ can substantially enhance FL for the large-scale system Linux kernel at a affordable cost.

In summary, by enhancing the capabilities of existing agents, LINUXFL⁺ facilitates more accurate FL with minimal costs. Our findings underscore the potential of LINUXFL⁺ to significantly support software maintenance tasks in Linux kernel.

### 5.3.2 METHOD-LEVEL FL

To further evaluate LINUXFL⁺ at a finer granularity, we extend our evaluation to method-level FL. Specifically, given the buggy files predicted by LINUXFL⁺, we proceed to identify buggy methods by prompting LLMs with a skeleton representation of each file, following prior work (Xia et al., 2024). This skeleton format preserves only function signatures and comments, which reduces input length while retaining essential context. The LLM is then prompted to identify the top-k (k=10) most relevant functions. Given the characteristics of the C language, we define method-level elements as functions, structures, and other code blocks. We consider the methods that are modified in the developer-committed patches as the ground truth for buggy methods.

Table 6 presents the method-level FL results of existing agents and those enhanced with LINUXFL⁺ based on GPT-4o. Overall, LINUXFL⁺ can consistently improve agents in method-level FL for Linux kernel. All three agent baselines exhibit low Recall@1 (below 0.1), while LINUXFL⁺ consis-

Table 6: Method-level FL results.

| Methods | Recall@1 | Recall@5 | Recall@10 | MRR |
|---|---|---|---|---|
| SWE-Agent | 0.089 | 0.178 | 0.214 | 0.170 |
| - w/ LINUXFL⁺ | 0.138 | 0.271 | 0.326 | 0.253 |
| AutoCodeRover | 0.042 | 0.088 | 0.094 | 0.077 |
| - w/ LINUXFL⁺ | 0.137 | 0.292 | 0.349 | 0.259 |
| Agentless | 0.098 | 0.147 | 0.179 | 0.162 |
| - w/ LINUXFL⁺ | 0.111 | 0.229 | 0.269 | 0.217 |

tently improves this metric beyond 0.1. The improvements are more pronounced in other metrics, e.g., for Recall@10, LINUXFL⁺ enhances all baselines by more than 0.09. While localizing finer-grained elements is inherently much more challenging specifically for large scale systems like Linux kernel, the overall accuracy at method level remains relatively lower than at the file level, highlighting the need for further research in this direction.

## 6 LIMITATIONS

**Limited Evaluation on Different LLMs.** To ensure consistency with prior work (Yang et al., 2024; Zhang et al., 2024; Xia et al., 2024) and facilitate fair comparison of agent performance across SWE-bench and LINUXFLBENCH, most experiments in this study employed GPT-4o as the backbone LLM. To address this limitation, we also validated the effectiveness of LINUXFL⁺ with open-source Qwen-32B. While LINUXFL⁺ consistently yields significant improvements, its performance with other LLMs was only briefly explored.

**Rough Usage of Mail Data.** LINUXFL⁺ leverages external knowledge from Linux kernel mailing list to enhance FL. Given the richness of email content, this resource may also contain irrelevant or outdated discussions, though it is valuable. To mitigate this, we employ various filtering and querying strategies, such as query reformulation and heuristic filtering, to improve the quality of retrieved emails. Despite these efforts, there is still room for further enhancement. Future work could explore more sophisticated approaches to effectively utilize mailing list knowledge for improved software maintenance tasks on the Linux kernel.

## 7 CONCLUSION

In this work, we introduce LINUXFLBENCH, a new and challenging software engineering benchmark designed for fault localization in the Linux kernel. To assess the effectiveness of existing LLM agents in complex software systems, we conduct an empirical study using LINUXFLBENCH. Initial results reveal that these agents struggle to accurately identify buggy files. To address this challenge, we propose LINUXFL⁺, a fault localization enhancement framework that leverages diverse expansion strategies to enrich candidate selection. Our approach demonstrates substantial improvements in localization performance.

## Reproducibility statement

We have taken steps to ensure the reproducibility of our results. All experimental settings are thoroughly described in the main text and appendix, and both the data and source code used in our work are made available at `https://anonymous.4open.science/r/LinuxFLBench-7C0D`.

## Ethics statement

All authors of this work have adhered to the ICLR Code of Ethics. Human involvement in this study was limited to the manual inspection step during the construction of our benchmark, LinuxFLBench. This task was reviewed and approved by the Institutional Review Board (IRB) at our institution. All participants were compensated at a rate of $15 per hour.

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

# A    APPENDIX

# B    ADDITIONAL DETAILS OF LINUXFLBENCH

## B.1    CONSTRUCTION PIPELINE OF LINUXFLBENCH

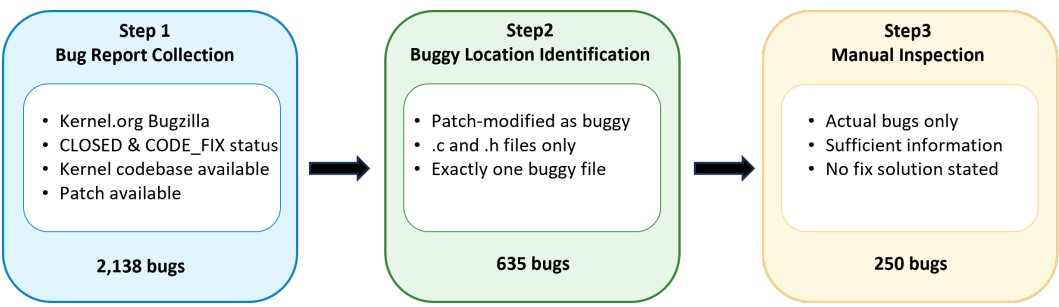

Figure 5: Construction pipeline of LINUXFLBENCH.

Fig. 5 illustrates the pipeline for constructing LINUXFLBENCH, which consists of three main phases: *Bug Report Collection*, *Buggy Location Identification*, and *Manual Inspection*. Through this process, we curate a total of 250 high-quality fault localization tasks.

## B.2    A SAMPLE KERNEL BUG FROM LINUXFLBENCH

We collect Linux kernel bugs (as shown in Fig. 6) from the reported and fixed bugs on Kernel.org Bugzilla. For each bug, the key information includes:

1. **Title:** The summary title of the bug report.

2. **Description:** A human-written description of the bug, which may include various types of information such as observed buggy behavior, reproduction steps in natural language, system logs, or call traces.

3. **Product:** The product category to which the bug is assigned.

4. **Component:** The specific component within the product affected by the bug.

5. **Hardware:** The hardware configuration on which the bug was observed.

6. **Kernel Version:** The version of the Linux kernel in which the bug occurred (e.g., 5.6.7).

7. **Paths:** The paths of the buggy files, extracted from the golden patch that fixes the bug.

8. **Modified Functions:** The method-level code elements modified by the patch.

## B.3    COMPARISON WITH SWE-BENCH LITE

To investigate the complexity and challenge of our benchmark, we conducted a series of quantitative analyses in comparison with SWE-bench Lite. The results are presented in Table 7 and summarized as follows.

**File length.** We compared the sizes of buggy files and patches (measured in lines of code). As shown in the table, LINUXFLBENCH contains substantially larger buggy files and patches, indicating higher complexity and greater difficulty for fault localization.

Figure 6: A sample kernel bug from LinuxFLBench.

Table 7: Comparison between LinuxFLBench and SWE-bench Lite

| Dataset | File Statistics (Lines) | | | | Location Information Type | | Stack Trace | Directory |
|---|---|---|---|---|---|---|---|---|
| | Mean Buggy | Mean Patch | Max Buggy | Max Patch | No Keywords | Exact Mention | Avg. Length | Avg. Size |
| LinuxFLBench | 2050.08 | 22.32 | 20142 | 572 | 0.568 | 0.008 | 14.33 | 16 files |
| SWE-bench Lite | 1211.13 | 10.13 | 8237 | 76 | 0.487 | 0.160 | 5.73 | 8 files |

**Directory size.** To indirectly capture the scope of potentially relevant files, we analyze the average number of files per directory. LinuxFLBench has an average of 16 files per directory, while SWE-bench Lite has only 8, suggesting that fault localization in our benchmark requires reasoning over larger and more interconnected contexts.

**Stack trace length.** Some bug reports in our benchmark include stack traces, which reflect the propagation paths of underlying bugs. On average, LinuxFLBench reports contain 14.33 functions per stack trace, compared to 5.73 in SWE-bench Lite. This suggests that bugs in our dataset involve longer propagation chains and more complex interactions.

**Location information.** Following the methodology of (Xia et al., 2024), we analyze the overlap between issue descriptions and file location information. Specifically, we distinguish between (i) straightforward bugs, where the full file path is explicitly mentioned in the description, and (ii) challenging bugs, where no related keywords appear. The results show that location information in LinuxFLBench is significantly sparser than in SWE-bench Lite, further increasing the difficulty of fault localization.

## C  Details of Baselines Used in This Paper

### C.1  Studied LLM agents.

This paper evaluates three SOTA LLM agents: SWE-Agent (Yang et al., 2024), AutoCodeRover (Zhang et al., 2024), and Agentless (Xia et al., 2024).

- **SWE-Agent.** SWE-Agent navigates the entire repository to identify the bug's location. To adapt this system to our benchmark, we modified the task description in the system prompt, specifying the objective as identifying suspicious files, while keeping the rest of the framework unchanged.

- **AutoCodeRover.** AutoCodeRover locates suspicious Python files based on the give GitHub issues through advanced code search techniques. We extended its functionality to support C/C++ projects by replacing its parser with *ctags*, enabling it to perform code search within Linux kernel codebases. Moreover, we also manually

sampled and inspected the trajectories of the agent to ensure proper handling of C language features.

- **Agentless.** Agentless identifies the suspicious files based on a concise representation of the repository structure. Given the vast number of files in the Linux kernel, we partition the repository structure into manageable portions by folder and feed them to the LLM in multiple iterations.

## C.2 IR-based Baselines.

To further investigate the effectiveness of agent-based methods, we also selected traditional IR-based baselines for comparison. Specifically, we included the classic methods BugLocator (Zhou et al., 2012) and BLUiR (Saha et al., 2013), along with widely used IR techniques such as BM25 (Robertson et al., 1995) and Sentence-BERT (Reimers & Gurevych, 2019).

- **BM25.** BM25 is one of the most widely used IR methods, and we include it as one of our baselines. BM25 is a bag-of-words retrieval function that ranks a set of documents based on term frequency and inverse document frequency of each document.

- **BugLocator.** BugLocator retrieves buggy files from a codebase by treating a bug report as a query and ranking files based on similarity using a revised vector space model (rVSM). The rVSM method prioritizes longer documents, assuming these files are more likely to contain bugs. Additionally, BugLocator incorporates historical bug fixes to further assess the likelihood of defects in a given file. In this work, we do not leverage this historical bug fix feature of BugLocator due to the unavailability of the necessary data.

- **BLUiR.** BLUiR enhances bug localization by extracting code entities, such as classes, methods, and variable names, from source code files. It calculates the relevance of these entities to both the title and description of a bug report respectively, aiding in the identification of buggy files.

- **SentenceBERT.** SentenceBERT enhances the traditional BERT model by incorporating siamese and triplet network architectures, enabling more efficient semantic search with reduced computational overhead. For our implementation, we utilize the sentence-transformer model *all-MiniLM-L6-v2*.

# D Failure Cases of LLM agents on LinuxFLBench

The suboptimal performance of agent-based methods can be attributed to several limitations, including confusion among related files and insufficient exploration of potential root causes. This section presents representative failure cases to illustrate these limitations.

## D.1 Confusion Among Related Files

An illustrative case is shown in Fig. 7. In this example, the update of the computer's battery and AC status involves interactions among the ACPI battery, AC adapter, and the embedded controller (EC). The corresponding drivers for these components all reside in the *drivers/acpi* directory. While different agent baselines identify the files related to the ACPI battery and adapter, they confuse and overlook the deeper component in the bug propagation chain—the EC driver—resulting in incorrect FL.

## D.2 Limited Exploration of Potential Causes

A representative case is provided in Fig. 8. the bug behavior "hangs on shutdown" could stem from various causes, since system shutdown involves a sequence of operations across

Figure 7: An illustrative case for "Confusion Among Related Files".

Figure 8: An illustrative case for "Limited Exploration of Potential Causes".

multiple components. Agents with limited exploration may employ a "depth-first search"-like strategy, focusing on superficially obvious reasons—such as failures in general power-off routines—while overlooking less apparent causes rooted in hardware state handling.

# E   PROMPT DESIGN OF LINUXFL⁺

## E.1   PROMPT TEMPLATES IN DIRECTORY-AWARE EXPANSION

LINUXFL⁺ re-selects related files within the same directories as the originally predicted files. Given the bug report("*bug information*") and the list of files("*candidate files*") in these directories, the LLM is instructed to select the relevant files using the following prompt.

**Prompt for Directory-Aware Expansion:** Please look through the following Linux kernel bug report and candidate files, and select a list of files that one would need to edit to fix the bug.

Here is the information about the bug:

### Linux kernel bug report ###

{*bug information*}

###

Based on the bug provided above, I will present a list of candidate files that may be relevant to the bug.

### Candidate files ###

{*candidate files*}

###

Please select files that are most likely to need modification to fix this bug.

Your response should be in the format of a list of file paths, and should be ordered by relevance in descending order. Please return at most 10 files.

### output example ###

['net/ipv6/proc.c', 'net/ipv6/netfilter/ip6_tables.c']

###

Please format your response strictly according to the format provided above without commentary.

## E.2 Prompt Templates in Potential Cause Expansion

In the phase of Potential Cause Expansion, LinuxFL$^+$ instructs the LLM to enumerate as many potential causes as possible using two approaches: direct hypothesis and mail-augmented hypothesis. The prompt for Mail-Augmented Hypothesis is presented below. Given the bug report ("*bug information*") and the retrieved emails ("*mail content*), the LLM is prompted to generate potential causes along with corresponding fix suggestions and the affected code files in a specified JSON format. The prompt for Direct Hypothesis is similar, but without including the retrieved email content.

**Prompt for Mail-Augmented Hypothesis:** Please review the following Linux kernel bug report, and then deduce the possible causes of the bug and provide corresponding code files and a potential fix. The bug is known to be related to the kernel code, and the fix should involve modifications to kernel code files.

Here is the information about the bug:

### Linux kernel bug report ###

{*bug information*}

###

To assist in your analysis, here are some emails retrieved using BM25 that may be relevant to the bug. Use them to inspire and identify additional possible causes:

### Mails ###

{*mail content*}

###

Based on the bug provided above, please output the possible causes, relevant code files, and solutions. Your response should follow the format below.

### Output example ###

[ { 'cause': 'A description of the potential cause of the bug.', 'code_file': 'Path of the code file that is most likely related to the bug.', 'fix_solution': 'A short description of the fix solution to apply in the code file.' }, ... ]

###

Please ensure the following:

- List as many causes as possible, ordered by relevance in descending order, with the most likely cause first.

- For each cause, list all relevant code files and their corresponding fixes, but only provide one code file and one fix per entry.

- The relevant code file is not necessarily the one causing the bug but should be a file where the bug can be fixed.

- The code file should be in the format of "net/ipv6/proc.c".

- Format your response strictly according to the format provided above without commentary.

## F  ABLATION STUDY

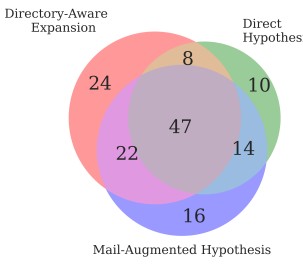

Figure 9: Venn Diagram for Correctly Localized Bugs by Agentless with Different Strategies.

Table 8: Evaluation results of LINUXFL$^+$ in different steps.

| Methods | Recall@1 | Recall@5 | Recall@10 | MRR |
|---|---|---|---|---|
| *Direct LLM Hypothesis* | 0.316 | 0.424 | 0.424 | 0.362 |
| SWE-Agent | | | | |
| *- w/ Directory-Aware Expansion* | 0.448 | 0.640 | 0.680 | 0.527 |
| *- w/ Direct Hypothesis* | 0.440 | 0.672 | 0.684 | 0.537 |
| *- w/ Mail-Augmented Hypothesis* | 0.488 | 0.632 | 0.632 | 0.549 |
| *- w/ Merge all* | 0.516 | 0.712 | 0.772 | 0.601 |
| AutoCodeRover | | | | |
| *- w/ Directory-Aware Expansion* | 0.424 | 0.592 | 0.608 | 0.492 |
| *- w/ Direct Hypothesis* | 0.444 | 0.636 | 0.636 | 0.528 |
| *- w/ Mail-Augmented Hypothesis* | 0.468 | 0.576 | 0.576 | 0.515 |
| *- w/ Merge all* | 0.476 | 0.692 | 0.74 | 0.570 |
| Agentless | | | | |
| *- w/ Directory-Aware Expansion* | 0.404 | 0.584 | 0.632 | 0.484 |
| *- w/ Direct Hypothesis* | 0.412 | 0.596 | 0.612 | 0.484 |
| *- w/ Mail-Augmented Hypothesis* | 0.396 | 0.520 | 0.520 | 0.447 |
| *- w/ Merge all* | 0.440 | 0.672 | 0.720 | 0.548 |

Table 8 presents the results of integrating individual components of LINUXFL$^+$ into the baselines based on GPT-4o, examining how each localization phase contributes to the final performance.

**Complementarity of Scaling Strategies.**  As shown in Table 8, agent baselines augmented with the three scaling strategies—Directory-Aware Expansion, Direct Hypothesis, and Mail-Augmented Hypothesis—exhibit varying FL performance on LINUXFLBENCH. Merging the results from these strategies leads to improved performance, suggesting their complementary nature. To further investigate this characteristic, we present a Venn diagram in Fig. 9, illustrating the top-1 successfully localized bugs achieved by each strategy beyond Agentless. Each strategy independently identifies a substantial number of bugs that the others fail to locate. This highlights the rationale behind our merging approach. The three strategies emphasize different aspects: directory-level structural information from the codebase, intrinsic knowledge from the LLM, and external expertise from historical mailing lists. Integrating these perspectives allows for more effective and robust fault localization.

**Effectiveness of Direct Hypothesis.**  The Direct Hypothesis strategy asks LLMs to directly infer buggy files from bug reports, independent of the outputs from agent-based methods. The results of this standalone approach, denoted as Direct LLM Hypothesis, are reported in the first row of Table 8. To further assess its effectiveness, we also evaluate its combination with agent baselines. Specifically, we integrate the predicted files from Direct Hypothesis with the original predictions of each agent, followed by a reranking step. As the results demonstrate, this strategy consistently improves localization performance across various agents. Although the standalone performance of Direct LLM Hypothesis is lower than that of the original agents, it provides complementary information that enriches both

(1) the original agent predictions (as shown in Table 8) and (2) other expansion strategies (as shown in Fig. 9). The primary goal of this strategy is to distill the internal knowledge of LLMs for understanding Linux kernel bugs. By integrating Direct Hypothesis with these agents and expansion strategies, we achieve a more robust and effective fault localization approach.

**Utility of Mail Retrieval.** As discussed in Section 6, LKML may contain irrelevant or outdated discussions. To evaluate our mail retrieval strategy, we first measure the proportion of retrieved emails that contain the correct buggy files. As shown in Table 9, our strategy significantly outperforms di-

Table 9: Mailing list retrieval analysis.

| Agent | Recall of Retrieved Mails | None → Found | Found → Lost |
|---|---|---|---|
| SWE-Agent | 0.536 | 0.128 | 0.080 |
| AutoCodeRover | 0.488 | 0.136 | 0.056 |
| Agentless | 0.460 | 0.132 | 0.116 |

rect BM25 retrieval (recall 0.332) on all agents, demonstrating its effectiveness. We further examine the impact on top-10 predictions under the Mail-Augmented Hypothesis by tracking two types of changes: (i) *None → Found*, where previously missing buggy files appear, and (ii) *Found → Lost*, where files drop out. The results indicate that expansion consistently adds correct files (e.g., 0.136 for AutoCodeRover) while rarely displacing existing ones, confirming that mail retrieval effectively enhances baseline

**Benefit of Mail Knowledge.** To investigate the benefits of incorporating knowledge from LKML, we compare baseline methods augmented with the Mail-Augmented Hypothesis against those only using the Direct Hypothesis. As shown in the Table 8, Mail-Augmented Hypothesis consistently outperforms Direct Hypothesis. The latter relies solely on the intrinsic knowledge of LLMs, without utilizing predictions from agent methods, and achieves a recall@1 of only 0.316. In contrast, with the assistance of mail knowledge, Mail-Augmented Hypothesis achieves a recall@1 as high as 0.488, with even more significant improvements observed in recall@10. These results demonstrate that mailing list data can effectively bridge the knowledge gap LLMs face in localizing bugs within the Linux kernel. It is worth noting that the effectiveness of Mail-Augmented Hypothesis varies across different agent methods. For instance, in the case of SWE-Agent, the predicted files facilitate the retrieval of more relevant emails, which provide stronger guidance during cause exploration.

**Impact of Re-Ranking.** LINUXFL$^+$ performs a re-ranking step on the candidates obtained from previous phase. The results in Table 8 demonstrate the effectiveness of this re-ranking process. By comparing the results after merging with the final performance of LINUXFL$^+$, further improvement in localization accuracy could be observed. It may stem from the high-quality candidate files provided by the different expansion strategies. With these enriched candidates, even a simple re-ranking allows the model to more easily identify the correct buggy files.

## G  HUMAN PARTICIPATION

In this work, human involvement is limited to the Manual Inspection step during the construction of our benchmark, LINUXFLBENCH. This task was approved by the Institutional Review Board (IRB) at our institution. All participants were compensated at a rate of $15 per hour.

During Manual Inspection, each annotator was provided with the following instruction: *"Given the title and description of the bug report, please label the report as 'yes,' 'no,' or 'unsure' for each of the following three questions: (1) Does the report describe an actual bug (e.g., not merely submitting a patch)? (2) Does the report contain sufficient information, such as clear natural language descriptions of the buggy behavior, reproduction steps, or detailed system logs? (3) Does the report avoid including solutions, such as identifying the buggy location or attaching patches? If unsure, please select the label 'unsure.'"* A report was assigned a final label of "yes" only if all three questions received a "yes" from an annotator. Each bug report was independently labeled by three participants. Reports that received at least two "yes" labels across annotators were retained in the final dataset.

# H   EXPERIMENT STATISTICAL SIGNIFICANCE

Table 10: Experiment Statistical Significance of LINUXFL$^+$

| Method | Enhanced (Mean ± Std) | Original (Mean ± Std) | Mean Diff | t-stat | p-value | CI (Enhanced) | CI (Original) |
|---|---|---|---|---|---|---|---|
| Agentless | 0.549 ± 0.431 | 0.419 ± 0.463 | 0.129 | 6.126 | 0.000 | [0.493, 0.600] | [0.361, 0.471] |
| AutoCodeRover | 0.589 ± 0.437 | 0.435 ± 0.469 | 0.154 | 5.825 | 0.000 | [0.537, 0.643] | [0.374, 0.493] |
| SWE-Agent | 0.610 ± 0.433 | 0.476 ± 0.463 | 0.134 | 5.679 | 0.000 | [0.561, 0.663] | [0.416, 0.533] |

To evaluate the effectiveness of the proposed LINUXFL$^+$, we performed statistical significance tests comparing the MRR scores of LLM agents enhanced with LINUXFL$^+$ to those of their original counterparts. As presented in Table 10, all enhancements introduced by LINUXFL$^+$ yield statistically significant improvements, with paired t-tests producing p-values below 0.0001. estimated using 1,000 resamples. Importantly, the confidence intervals for the enhanced models do not overlap with those of the original models, providing additional evidence for the significance of the observed improvements. These consistent and statistically significant gains across multiple LLM agents underscore the robustness and effectiveness of our FL-enhancing framework.

# I   LLM USAGE

In preparing this work, we used LLMs as an assistive tool. Specifically, LLMs (e.g., ChatGPT) were employed to refine the clarity and readability of manuscript drafts through language polishing. Importantly, all research ideas, methodology design, experimental implementation, and analysis were conceived and conducted by the authors. The LLMs were not used for generating research hypotheses, designing experiments, or interpreting results. The authors take full responsibility for the content of this paper.

