# OpenReview forum: "Benchmarking and Enhancing LLM Agents in Localizing Linux Kernel Bugs"
_ICLR.cc/2026/Conference — ICLR 2026 Conference Withdrawn Submission_

### Official Review · Reviewer_cTfW · 2025-10-24

**Soundness:** 3
**Presentation:** 4
**Contribution:** 2
**Rating:** 2
**Confidence:** 4

**Summary:**

This paper introduces LinuxFLBench, a fault localization benchmark for Linux that includes 250 manually verified cases. Compared with SWE-Bench, LinuxFLBench presents more challenging Linux bug reports, primarily due to the large scale of the Linux codebase and the diversity of bug causes. In the evaluation section, the authors assess both traditional information retrieval–based methods and agent-based fault localization methods, finding that compared with SWE-Bench, these agents experience more than a 15% drop in recall on LinuxFLBench. Finally, the authors propose a new framework to mitigate these challenges.

**Strengths:**

- The paper is well-written and clearly presented.
- The proposed benchmark introduces a higher level of difficulty compared to SWE-Bench.
- This benchmark focuses on user-reported bugs rather than fuzzing-based ones, which makes the dataset more realistic.
- The experimental evaluation is fairly comprehensive, covering most mainstream fault localization approaches.

**Weaknesses:**

- When constructing the benchmark, the authors still define the ground-truth root cause based on the patch location. While simple, this approach can be imprecise since the patch location does not always correspond to the actual root cause.
- The benchmark focuses solely on Linux bugs. Although Linux is one of the largest codebases, the motivation provided by the authors is not fully convincing:
  - **Limited Observability**: the main challenge lies in the lack of runtime information. However, couldn’t one instead use bug reports from other large projects such as Redis or QEMU, filtering out those that include runtime information? Moreover, this challenge does not seem to be reflected in the benchmark’s evaluation results. The difficulties summarized in the evaluation are also unrelated to this point.
  - **Diverse Impact**: while the diversity of root causes is indeed a major challenge in FL, is this a sufficient reason to choose only Linux? Couldn’t other large-scale projects (e.g., Redis, QEMU) also achieve this? Furthermore, does maximizing root-cause diversity conflict with the decision to focus solely on Linux cases?
- In the evaluation, the authors only used GPT-4o, which is not a state-of-the-art (SOTA) LLM and lacks strong reasoning capabilities. Since the performance of agents is highly correlated with model capability, although the authors discuss this limitation, I would still encourage them to include comparisons with current SOTA models such as GPT-5 or Claude 4.5.

**Questions:**

- When evaluating these agents, did the authors provide them with an execution environment?
- Were the agents’ tool invocation or command execution abilities (e.g., writing and running search scripts during iteration) considered in the experiments?

---

> ### Author Response · Authors · 2025-11-22
>
> Firstly, we would like to thank the reviewer for putting in time and effort to provide feedback.
>
> **1: “When evaluating agents, did the authors provide an execution environment?”**
>
> We provided a controlled execution environment for all agents, enabling command execution, file inspection, and repository interaction during localization, consistent with their standard usage. We did not provide full kernel execution or test-triggering support, since user-reported bugs typically lack runnable environments/tests. Enabling full kernel execution would require substantial additional infrastructure beyond the scope of fault localization and is left for future work.
>
> **2: “Were tool invocation or command execution abilities (e.g., writing/running search scripts) allowed?”**
>
> Yes. We enabled tool invocation and command execution. Moreover, to ensure that agents functioned properly in a C codebase, we extended their capabilities to support C projects—for example, by adapting AutoCodeRover’s search functionality through replacing its parser. We also manually sampled and inspected agent trajectories to confirm correct handling of C-specific language features and tool usage.
>
> **3: “Patch location does not always correspond to root cause—how do the authors justify using patches as ground truth?”**
>
> While we acknowledge patch locations may not always perfectly align with the semantic root causes, using patch-modified regions as ground-truth localization targets is a widely adopted and practical approximation in prior FL research [1,2,3]. For example, Defects4J—one of the most widely used FL benchmarks with more than 1,800 citations—follows the same assumption when defining ground-truth locations. To reduce ambiguity, we retain only cases where the patch unambiguously identifies a single faulty file. This yields a reliable and reproducible labels. We welcome suggestions from the reviewer regarding principled alternatives for large-scale root-cause annotation, which remains a challenging open problem.
>
> **4: “Why focus only on Linux? Could other large projects (e.g., Redis, QEMU) also satisfy the stated motivations?”**
>
> - *Why Linux?* We focus on Linux not only because it is highly challenging, but also because it is critically important. (1) The Linux kernel is a long-standing, large-scale, and influential real-world software project (28M+ LOC, 20k+ contributors [4]) with rich community artifacts like mailing lists, making it an ideal FL benchmark target. (2) There is also strong precedent in the literature for developing quality assurance techniques [5][6][7] and benchmarks [8][9] specifically for the Linux kernel, further underscoring its importance. We believe that building automated FL techniques tailored for this domain can bring substantial benefits to the broader developer community.
>
> - *Could other large projects also satisfy the stated motivations?* We agree Redis/QEMU are valuable future targets, but our goal here is to expose the gap between existing benchmarks (e.g., SWE-bench) and much more complex real-world systems like the kernel. In fact, based on our findings that reveal the limited capabilities of existing agents in localizing kernel bugs, we believe it is actually necessary and beneficial for the community to invest more effort in assessing and enhancing agents’ ability to tackle challenging tasks in complex, real-world systems. We view LinuxFLBench as a first step toward broadening the evaluation landscape, and we encourage future extensions to additional complex and semantically demanding systems such as Redis or QEMU.
>
> **5: “Comparisons with current SOTA models such as GPT-5 or Claude 4.5.”**
>
> We additionally evaluated Agentless with GPT-5 on 50 sampled issues. Even GPT-5 remains challenged on kernel bugs, confirming the difficulty of FL in complex systems. LinuxFL+ still delivers substantial gains, showing it complements stronger backbones.
>
> |   | Recall@1 | Recall@5 | Recall@10 |   MRR   |
> |-----------|----------|----------|-----------|---------|
> | Agentless (GPT-5) | 0.480 | 0.660 | 0.760 | 0.569 |
> | LinuxFL+ (GPT-5)  | 0.540 | 0.880 | 0.880 | 0.668 |
>
> ---
> References
>
> [1] Just et al., Defects4J: a database of existing faults to enable controlled testing studies for Java programs, ISSTA 2014.
>
> [2] Chen et al., LocAgent: Graph-Guided LLM Agents for Code Localization, ACL 2025.
>
> [3] Xia et al., Agentless: Demystifying LLM-based software engineering agents, 2024.
>
> [4] Linux Foundation, Linux Kernel History Report, 2020.
>
> [5] Yang et al., KernelGPT: Enhanced kernel fuzzing via large language models, ASPLOS 2025.
>
> [6] Liu et al., Understanding the Linux kernel, visually, EuroSys 2025.
>
> [7] Xu et al., Concurrency Testing in the Linux Kernel via eBPF, arXiv 2025.
>
> [8] Mathai et al., KGym: A platform and dataset to benchmark LLMs on Linux kernel crash resolution, NeurIPS 2024.
>
> [9] Borges et al., Linux Kernel Configurations at Scale: A Dataset for Performance and Evolution Analysis, arXiv 2025.

---

> > ### Comment · Reviewer_cTfW · 2025-11-27
> >
> > Thanks for the response, which helps clarify my understanding. But I still have concerns regarding benchmark quality and project selection.
> >
> > While using diff patches or commit messages for fault localization is a common practice, I believe that when introducing a benchmark, the reliability of the ground truth should be the highest priority. For example, in Defects4J, each case is fully reproducible, which ensures its correctness and reliability. In your benchmark, you mention manually filtering out low-quality issues. In that case, would it also be possible to manually verify the root cause of each issue to ensure the reliability of the ground truth?
> >
> > I agree that Linux is a large-scale, complex, and well-profiled system. But the author's motivation for focusing on Linux only is still not convincing enough.
> >
> > For now, I will maintain my score.

---

> > > ### Author Response · Authors · 2025-11-28
> > >
> > > We thank the reviewer cTfW’s  acknowledgement that our response has clarified the reviewer’s understanding. We then further respond to the two remaining concerns (benchmark quality and project selection) mentioned in the comments.
> > >
> > > **Benchmark Quality.**
> > >
> > > Though it has been the common practice of using developer-patch as ground-truth in fault localization benchmarks (including Defects4J,  BugLocator[1], BLUiR[2,3], LOC-Bench[4], and many others[5,6]), we’d like to clarify the additional effort we’ve made to maximize the reliability of our benchmark:
> > >
> > > - Trusted data source.  Different from casual open-source projects, all patches in LinuxFLBench come from the official Linux kernel development workflow: they were discussed, reviewed, and accepted by kernel maintainers before being merged upstream. Kernel patches undergo strict community review, so the modified locations reliably reflect the developer-verified fix.
> > >
> > > - Ambiguity reduction through single-file selection. We keep only cases where the maintainer-accepted patch modifies exactly one file. Multi-file patches often include auxiliary or refactoring edits that do not correspond to the true root cause, so restricting to single-file fixes ensures clean and unambiguous ground truth.
> > >
> > > Together, these measures provide the highest-fidelity ground truth that is realistically obtainable for large-scale, real-world Linux bugs
> > >
> > > **The motivation of selecting Linux kernel.**
> > >
> > > -  As mentioned in our previous response, we choose Linux given both its challenge and importance (a long-standing, large-scale, and influential real-world software project with 28M+ LOC, 20k+ contributors).
> > > - Moreover, quality assurance specifically for the Linux kernel has already become a widely-studied research topic ([7][8][9][10][11]).
> > > - Lastly, and most importantly, our work aims to raise the community’s awareness of the substantial gap between debugging agents that handle relatively-simple software (e.g., the library-style subjects in SWE-bench) and those needed for far more complex, real-world systems such as kernels. We position LinuxFLBench as an initial step toward broadening the evaluation landscape and hope it will inspire further research on building debugging agents for other complex software systems
> > >
> > > ---
> > >
> > > References
> > >
> > > [1] J. Zhou, H. Zhang and D. Lo, "Where should the bugs be fixed? More accurate information retrieval-based bug localization based on bug reports," 2012 34th International Conference on Software Engineering (ICSE), Zurich, Switzerland, 2012, pp. 14-24, doi: 10.1109/ICSE.2012.6227210.
> > >
> > > [2] R. K. Saha, J. Lawall, S. Khurshid and D. E. Perry, "On the Effectiveness of Information Retrieval Based Bug Localization for C Programs," 2014 IEEE International Conference on Software Maintenance and Evolution, Victoria, BC, Canada, 2014, pp. 161-170, doi: 10.1109/ICSME.2014.38.
> > >
> > > [3] R. K. Saha, M. Lease, S. Khurshid and D. E. Perry, "Improving bug localization using structured information retrieval," 2013 28th IEEE/ACM International Conference on Automated Software Engineering (ASE), Silicon Valley, CA, USA, 2013, pp. 345-355, doi: 10.1109/ASE.2013.6693093.
> > >
> > > [4] Zhaoling Chen, Robert Tang, Gangda Deng, Fang Wu, Jialong Wu, Zhiwei Jiang, Viktor Prasanna, Arman Cohan, and Xingyao Wang. 2025. LocAgent: Graph-Guided LLM Agents for Code Localization. In Proceedings of the 63rd Annual Meeting of the Association for Computational Linguistics (Volume 1: Long Papers), pages 8697–8727, Vienna, Austria. Association for Computational Linguistics.
> > >
> > > [5] Kang S, An G, Yoo S. A quantitative and qualitative evaluation of LLM-based explainable fault localization[J]. Proceedings of the ACM on Software Engineering, 2024, 1(FSE): 1424-1446.
> > >
> > > [6] Chang J, Zhou X, Wang L, et al. Bridging Bug Localization and Issue Fixing: A Hierarchical Localization Framework Leveraging Large Language Models[J]. arXiv preprint arXiv:2502.15292, 2025.
> > >
> > > [7] Yang et al. Kernelgpt: Enhanced kernel fuzzing via large language models[C]//Proceedings of the 30th ACM International Conference on Architectural Support for Programming Languages and Operating Systems, Volume 2. 2025: 560-573.
> > >
> > > [8] Liu H, Jiang Y, Xu C. Understanding the linux kernel, visually[C]//Proceedings of the Twentieth European Conference on Computer Systems. 2025: 1044-1060.
> > >
> > > [9] Xu J, Wolff D, Han X Y, et al. Concurrency Testing in the Linux Kernel via eBPF[J]. arXiv preprint arXiv:2504.21394, 2025.
> > >
> > > [10] Mathai A, Huang C, Maniatis P, et al. Kgym: A platform and dataset to benchmark large language models on linux kernel crash resolution[J]. Advances in Neural Information Processing Systems, 2024, 37: 78053
> > >
> > > [11] Borges H, Pereira J A, Khelladi D E, et al. Linux Kernel Configurations at Scale: A Dataset for Performance and Evolution Analysis[J]. arXiv preprint arXiv:2505.07487, 2025.-78078.

---

### Official Review · Reviewer_3H9u · 2025-10-30

**Soundness:** 3
**Presentation:** 2
**Contribution:** 3
**Rating:** 6
**Confidence:** 3

**Summary:**

This paper introduces LinuxFLBench, a benchmark of 250 real-world Linux kernel fault-localization (FL) tasks constructed from user-reported bugs (Kernel.org Bugzilla), spanning 120 kernel versions and 66 components. Each task provides a bug report, the buggy codebase, and ground-truth buggy file/method(s) derived from patches. The paper evaluates state-of-the-art LLM agents (SWE-Agent, AutoCodeRover, Agentless) and classical IR baselines (BM25, BugLocator, BLUiR, Sentence-BERT). Results show that LLM agents outperform classical IR methods but still struggle on the Linux kernel compared to general software benchmarks like SWE-bench (e.g., best top-1 file-level recall is 41.6% vs. ~70% on SWE-bench). To address these gaps, the authors propose LinuxFL+, a post-hoc enhancement framework with (i) directory-aware expansion, (ii) potential-cause expansion via LLM hypothesis (direct and mail-augmented using LKML), and (iii) candidate integration with re-ranking. LinuxFL+ improves all agents with modest token cost (e.g., SWE-Agent Recall@10: 0.584 → 0.768; MRR: 0.476 → 0.610), and provides consistent gains even for a smaller open-source backbone (Qwen3-32B). Method-level FL remains challenging (Recall@1 < 0.14 even with LinuxFL+).

**Strengths:**

- **Originality**: New, domain-specific FL benchmark for Linux with real user-reported bugs at larger scale than SWE-bench(-lite). Novel and important observation of degraded agent performance on the kernel.
- **Quality**: Careful curation; strong baseline coverage (IR and agents); robust gains from LinuxFL+ across agents/backbones; statistical significance analysis; cost accounting shows practical overhead (~$0.04/task).
- **Clarity**: Clear motivation and high-level method; results/ablations are informative; limitations stated.
- **Significance**: Highlights challenges of applying LLM agents to large, low-observability, multi-factor systems. Likely practical impact for practitioners evaluating Linux bugs.

**Weaknesses:**

- **Benchmark bias via single-file constraint**: Excluding multi-file fixes may skew composition and agent failure modes. Please quantify exclusions, characterize them, and consider a multi-file track.
- **Leakage/provenance in mail-augmentation**: Despite pre-report restriction and filters, subtle leakage is possible. A formal audit (time-window sensitivity, near-duplicate detection, blinded checks) and a “clean” subset would strengthen claims.
- **Heuristic under-specification**: Directory-aware expansion (choice of k), merging and LLM re-ranking are lightly formalized. Compare against strong non-LLM re-rankers (BM25/SPLADE/ColBERT on path+report) and provide sensitivity studies.
- **Limited LLM/agent diversity**: Most results rely on GPT-4o with one open-source model. Explore more open-source backbones and alternative agent frameworks/policies to bolster generality.
- **Method-level FL still weak**: Although improved, recalls remain low. Decompose errors into file recall vs. intra-file ranking; explore alternative skeletonization (call-graph stubs, macro expansion hints).
- **Scope/portability**: C/Linux focus. Discuss portability to other large C/C++ systems and consider releasing scripts to replicate the construction elsewhere.

**Questions:**

1. **Leakage auditing**: Can you provide a formal audit for mail-augmentation (time windows, near-duplicate detection, blinded assessment that emails don’t directly reveal locations)?
2. **Single-file constraint**: What fraction of reports were excluded due to multi-file patches? How do agents and LinuxFL+ perform on a multi-file track?
3. **Sensitivity analyses**: Report sensitivity to directory expansion k, retrieval top-k, and re-ranking. Are gains robust across settings?
4. **Non-LLM re-ranking baselines**: How does LLM re-ranking compare to strong lexical/sparse/dense rankers over file path + report?
5. **Backbone diversity**: Beyond GPT-4o and Qwen3-32B, how do other open-source models (7B–72B) fare? Any scaling trend?
6. **Method-level bottlenecks**: Can you decompose failure into (i) file recall vs. (ii) method ranking? Would richer skeletons help?
7. **Generalization**: Any early results on other kernels/large C++ codebases? Will you release scripts for reproducing analogous benchmarks?
8. **Cost/latency**: Per-task wall-clock times with/without LinuxFL+ and guidance on batching/parallelization?

---

> ### Author Response · Authors · 2025-11-22
>
> **1: “Audit for mail-augmentation”**
>
> Data leakage has been carefully addressed in our mail-augmentation design. We (i) remove emails with external URLs or explicit “Bugzilla” mentions, and (ii) enforce strict **time-window filtering** (emails strictly pre-dating the report). These prevent mail augmentation from leaking fix locations.
>
>
> **2: “Fraction of multi-file bugs”**
>
> Among valid-patch bugs we collected, 61.28% are single-file and 38.72% multi-file. We currently include only single-file cases to ensure clean ground truth. We have not yet evaluated a multi-file track because reliable labeling requires substantial manual effort, but we plan to build and test a curated multi-file extension in future work.
>
> **3: “Sensitivity studies across settings**
>
> We ran sensitivity analyses (50 sampled issues) of LinuxFL+ based on Agentless. Gains remain stable across settings.
>
> *Directory-aware expansion depth (k ∈ {1,5,10} files expanded):*
>
> | Directory k | Recall@1 | Recall@5 | Recall@10 |  MRR  |
> |-------------|----------|----------|-----------|-------|
> | Agentless   | 0.280    | 0.380    | 0.420     | 0.324 |
> | Top-1       | 0.280    | 0.440    | 0.480     | 0.349 |
> | Top-5       | 0.300    | 0.420    | 0.420     | 0.357 |
> | Top-10      | 0.320    | 0.480    | 0.520     | 0.389 |
>
> *Mail retrieval depth for Potential Cause Expansion (top-k mails ∈ {1,5,10}):*
>
> | Retrieval top-k mails | Recall@1 | Recall@5 | Recall@10 |  MRR  |
> |-----------------------|----------|----------|-----------|-------|
> | Agentless             | 0.280    | 0.380    | 0.420     | 0.324 |
> | Top-1                 | 0.380    | 0.560    | 0.560     | 0.455 |
> | Top-5                 | 0.400    | 0.500    | 0.500     | 0.450 |
> | Top-10                | 0.360    | 0.580    | 0.580     | 0.450 |
>
> *Re-ranking depth (k ∈ {3,5,10} candidates; no need for k=1):*
>
> | Re-ranking k | Recall@1 | Recall@5 | Recall@10 |  MRR  |
> |--------------|----------|----------|-----------|-------|
> | Original     | 0.440    | 0.672    | 0.720     | 0.548 |
> | Top-3        | 0.440    | 0.620    | 0.620     | 0.527 |
> | Top-5        | 0.440    | 0.660    | 0.660     | 0.532 |
> | Top-10       | 0.440    | 0.684    | 0.724     | 0.549 |
>
> **4: “LLM re-ranking vs strong non-LLM rankers?”**
>
> We replaced the LLM re-ranker with BM25 over file paths. BM25 re-ranking is consistently weaker than GPT-4o:
>
> | Agent         | Re-ranker | Recall@1 |
> |---------------|-----------|----------|
> | Agentless     | GPT-4o    | 0.440    |
> |               | BM25-path | 0.236    |
> | AutoCodeRover | GPT-4o    | 0.500    |
> |               | BM25-path | 0.296    |
> | SWE-agent     | GPT-4o    | 0.524    |
> |               | BM25-path | 0.288    |
>
>
> BM25 struggles because (i) we remove explicit path-revealing reports, and (ii) kernel reports may be noisy with weak lexical overlap, while LLMs leverage semantics.
>
> **5: “More open-source backbones / scaling trend?”**
> On 50 tasks with Agentless + LinuxFL+, Qwen3-8B/14B/32B show consistent gains and a genenral scaling trend:
>
> | Model     | R@1 | R@5 | R@10 | MRR |
> |-----------|-----|-----|------|-----|
> | Agentless | 0.280 | 0.380 | 0.420 | 0.324 |
> | Qwen3-8B  | 0.300 | 0.500 | 0.540 | 0.386 |
> | Qwen3-14B | 0.320 | 0.480 | 0.480 | 0.383 |
> | Qwen3-32B | 0.400 | 0.560 | 0.640 | 0.476 |
> | GPT-4o    | 0.460 | 0.680 | 0.740 | 0.556 |
>
> **6: “Decompose method-level failures / richer skeletons?”**
>
> *Failure decomposition (MethodRecall@1)*
>
> File recall is the dominant bottleneck. Conditional method accuracy remains low, so method-level FL still needs work:
>
> | Agent         | File-recall fail (F@1=0) | Method-rank fail (F@1=1, M@1=0) | M@1 \| F@1 |
> |---------------|----------------------------|-----------------------------------|-----------|
> | Agentless     | 56.00%                     | 30.40%                            | 0.309     |
> | AutoCodeRover | 50.00%                     | 32.00%                            | 0.360     |
> | SWE-agent     | 47.60%                     | 34.40%                            | 0.344     |
>
> *Richer skeletons*
> We tried call-graph context, but kernel call-graph extraction is costly and suffers path explosion. We keep lightweight skeletons (signatures) and will explore richer, efficient skeletons (macro hints, dependency summaries) in future work.
>
> **7: “Other codebases / release scripts?”**
>
> LinuxFL+ is conceptually general. On 50 sampled SWE-bench (Python) issues, Directory-Aware and Potential Cause Expansions consistently improve Agentless (Recall@1 +0.040 / +0.220), suggesting good portability. We will release full data-construction scripts to support reproducibility and analogous benchmark building.
>
>
> **8: “Wall-clock time / batching?”**
> Base Agentless averages **20.18 LLM calls/task**, while LinuxFL+ adds ~**5 extra calls**, a modest overhead( ~**7.03s/task**). The three expansions are parallelizable within a task, and LLM calls can be batched across tasks; moderate parallelization brings per-task time close to the base agent.

---

### Official Review · Reviewer_bR7D · 2025-11-01

**Soundness:** 3
**Presentation:** 3
**Contribution:** 3
**Rating:** 6
**Confidence:** 3

**Summary:**

The paper tackles the problem of fault localization for large scale software systems like the linux kernel. The authors first propose a well-defined FL benchmark created from bug reports/patches in the Linux Kernel. The authors compare baseline agent-based/and other prior software issue solving techniques and demonstrate significant drop in fault localization on the benchmark. Furthermore the authors propose LinuxFL to improve localization performance specific on the linux kernel through a combination of reranking and retrievel using linux mailing list.

**Strengths:**

- deals with a very important problem of fault localization in large software systems such as the Linux kernel
- construct a benchmark for this specific task that can be used by future research projects
- benchmark includes manual annotation which ensures high quality

**Weaknesses:**

- benchmark ignores more complex and difficult bugs/fault localization tasks:
	- The authors claims that "we kept only unambiguous cases where exactly one file was modified to ensured the reliability of the ground truth". This would ignore a lot of multi-file bugs which are extremely important to evaluate
- baseline setups:
	- The paper compares against some prior baselines used for solving software development issues like the ones in SWE-bench.
	- however, the task in this work is different (fault localization), its unclear how the authors modify an approach (e.g., SWE-agent) for a fault localization task?
- proposed approach are aimed specifically for Linux kernels and may not generalize to other large systems:
	- building on my prior point regarding the baseline setup, the prior tools work well as they can be utilized for other repos as well as different software engineering tasks (e.g., repair, feature implementation)
	- On the other hand, the proposed LinuxFL+ is extremely tailored for Linux kernels with the authors even using Linux mailing lists

**Questions:**

1. How did the authors modify the prior agents to perform fault localization tasks instead of solving software issues
2. Please comment on if LinuxFL+ can be utilized for effective fault localization in other software systems. For example, have the authors applied similar approach for fault localization on the SWE-bench problems?
3. Using the Linux mailing list could lead to some data leakage (even excluding Bugzilla), as the developer may discuss patches that are very relevant for the bug you want to localize, how did the authors address this issue?
4. Can the authors also comment on if the constructed benchmark can be utilized for repair evaluation?

---

> ### Author Response · Authors · 2025-11-22
>
> Firstly, we would like to thank the reviewer for putting in time and effort to provide feedback.
>
> **1: “How did the authors modify prior agents (e.g., SWE-agent) to perform fault localization instead of issue solving?”**
>
> For agents such as AutoCodeRover and Agentless, whose issue-solving workflows already include an explicit fault-localization stage, we simply retained their original pipelines but terminated the process once the agents produced the predicted buggy locations. For SWE-agent, which is a general-purpose software engineering agent capable of following task-specific instructions, we adapted it by just specifying the objective as identifying suspicious files directly in the prompt while leaving the remainder of its framework unchanged.
>
> **2: “Please comment on whether LinuxFL+ can be used for fault localization in other software systems (e.g., SWE-bench problems).”**
>
> We believe the *core idea* behind LINUXFL+ (i.e.,  enhancing LLM-based agents by integrating external domain knowledge) is general and should be adaptable to other domains. In other ecosystems, comparable external resources (e.g., GitHub Discussions, Stack Overflow threads, or project-specific documentation) could be leveraged similarly as LKML. Therefore, we believe our high-level ideas should be general and could future work  towards building powerful debugging agents for complex systems.
>
> To validate generality, we ran preliminary experiments on SWE-bench (50 sampled issues) by applying two LinuxFL+ strategies on top of Agentless. Since SWE-bench projects lack mailing lists, we use historical GitHub commits for Potential Cause Expansion. The results show consistent performance improvements when Agentless is enhanced with either of our two expansion strategies on SWE-bench, indicating the potential generality of our framework.
>
> | Strategy                     | Recall@1 | Recall@5 | Recall@10 | MRR   |
> |-----------------------------|----------|----------|-----------|--------|
> | Directory-Aware Expansion   | +0.040   | +0.080   | +0.080    | +0.060 |
> | Potential Cause Expansion   | +0.220   | +0.360   | +0.380    | +0.271 |
>
>
>
> **3: “Using the Linux mailing list could lead to data leakage … how did the authors address this?”**
>
> Data leakage has been carefully addressed in our use of the Linux kernel mailing list. Specifically, we have taken several measures to minimize this risk. During the mail-collection stage, we filtered out emails containing external URLs or references to “bugzilla,” which often indicate discussions involving known bugs or fixes. During retrieval, we further restricted candidate emails to those sent strictly *before* the corresponding bug report to maintain temporal consistency and avoid inadvertently using information that postdates the report. These safeguards ensure that mail augmentation does not access content revealing the actual fix locations.
>
> **4: “Can the constructed benchmark also be used for repair evaluation?”**
>
> Our dataset indeed includes the human-authored patches that repaired the reported bugs, which can in principle serve as ground-truth references for repair evaluation. Static similarity metrics—such as exact match or CodeBLEU—can be directly applied to assess patch correctness. But test-triggered evaluation is not currently supported, as user-reported bugs typically lack runnable environments or failure-triggering scripts. Providing the additional infrastructure needed for rigorous, execution-based repair evaluation is also an important direction for our future work.
>
> **5: “benchmark ignores more complex and difficult bugs/fault localization tasks”**
>
> Our benchmark currently focuses on bugs involving exactly one buggy file. The primary reason is to avoid labeling noise: when a patch modifies multiple files, not all edited files necessarily correspond to the true root cause, and treating all of them as “buggy” would introduce ambiguity and harm evaluation reliability.
> In fact, this setting is also common in practice and aligns with widely used Software Engineering benchmarks such as SWE-bench Lite[1], on which LLM agents like SWE-Agent, AutoCodeRover, and Agentless are typically evaluated.
> In addition, even under this single-file setting, our experiments already show that LLM-based agents face substantial challenges. While we acknowledge that this constraint may limit task complexity, we believe this work can serve as a necessary and reasonable starting point given the intricacy of the Linux kernel.
> Expanding the benchmark to include multi-file or cross-module cases is indeed valuable and is one of our key directions for future work. We plan to (i) release a multi-file track with carefully curated ground-truth file sets, and (ii) incrementally scale the benchmark over time by incorporating newly collected real-world bugs. We appreciate the reviewer’s suggestion.
>
> ---
> References:
>
> [1] SWE-Bench: Can Language Models Resolve Real-World Github Issue? Jimenez et al. ICLR 2024.

---

### Official Review · Reviewer_CisJ · 2025-11-01

**Soundness:** 1
**Presentation:** 3
**Contribution:** 2
**Rating:** 2
**Confidence:** 4

**Summary:**

This paper introduces LinuxFLBench, a new benchmark for evaluating fault localization (FL) in the Linux kernel, and proposes LinuxFL+, an enhancement framework designed to improve existing LLM-based agents’ performance on this challenging domain.
The authors benchmark several leading LLM agents (SWE-Agent, AutoCodeRover, Agentless) and find that they perform significantly worse on Linux kernel bugs than on existing datasets like SWE-bench.

**Strengths:**

- The paper tackles an important and underexplored domain, namely fault localization in large-scale, low-level software like the Linux kernel.

- Provides a well-constructed benchmark (LinuxFLBench) with real-world bug reports and ground-truth patches.

- The enhancement framework (LinuxFL+) offers practical, empirically validated gains with low cost and good reproducibility.

**Weaknesses:**

1. **Limited methodological novelty in LinuxFL+**
Although LinuxFL+ improves empirical results, the method primarily combines straightforward retrieval expansion and RAG-style augmentation.
The two strategies( i.e., directory-aware expansion and potential-cause expansion) are intuitive extensions rather than novel algorithmic contributions.
No clear theoretical insight or principled reasoning is provided for why these expansions work or how they might generalize beyond this specific setup.

2. **Benchmark coverage and representativeness remain narrow**
LinuxFLBench contains only 250 bugs despite the vast number of real-world Linux kernel issues.
Moreover, the dataset filters only “single-file fix” bugs, which simplifies the problem but undermines realism.
As a result, the benchmark might not capture the complexity of multi-file interactions and cross-component dependencies typical of real kernel debugging.

3. **Insufficient Evaluation analysis**
The evaluation remains superficial, reporting results without diagnostic analysis of why LLM agents fail or how the solution addresses specific reasoning errors. The failure analysis in Section 4.3 is anecdotal, lacking a systematic categorization of error types. Furthermore, the reasons behind the limited method-level improvements

4. **Missing discussion on contamination and data leakage**
Since LinuxFLBench is built from public Bugzilla and mailing list data, and the models (especially GPT-4o) are trained on internet-scale corpora, there is a non-trivial risk of data contamination.
The paper does not describe any checks for overlapping content between pretraining data and evaluation tasks.

**Questions:**

Q1: Could you clarify what key conceptual or methodological innovation distinguishes LinuxFL+ from standard RAG or search expansion frameworks?

Q2: LinuxFLBench includes only 250 single-file bugs, which may limit its realism and generalizability. How do you plan to address the limited dataset scale and the exclusion of multi-file or cross-module bugs?

Q3: Can you provide a more systematic analysis of the types of failures or reasoning gaps observed in LLM agents, and evidence or procedures confirming that LinuxFLBench and the LKML data are not contaminated with model pretraining data?

---

> ### Author Response · Authors · 2025-11-22
>
> **1: “Key conceptual or methodological innovation distinguishes LinuxFL+ from standard RAG or search expansion frameworks”**
>
> While standard RAG and search expansion are high-level generic ideas, LinuxFL+ introduces FL-specific enhancements for large-scale complex systems.
>
> Compared to standard RAG, LinuxFL+ redesigns retrieval with two domain-grounded improvements: (1) Retrieval space refinement: restricting the search space to emails associated with files predicted by the agent;  (2) Retrieval query reformulation: rewriting noisy bug reports into structured information. As reported in line 1039, this strategy consistently outperforms native BM25 (recall 0.332), achieving >0.46 recall across agents.
>
> Compared to generic search expansion, LinuxFL+ makes the first effort to show two fine-grained and unique strategies:(i) *directory-aware expansion*, broadening the search scope within structurally relevant directories of the initial predictions, and (ii) *potential-cause expansion*, using LLM-generated failure hypotheses (optionally mail-augmented) to surface semantically plausible root causes.
>
> Beyond LinuxFL+, we release **LinuxFLBench** from real kernel bugs and provide the empirical study showing that existing agents struggle on this complex domain, highlighting the need for improved FL on large real-world systems.
>
> **2: “LinuxFLBench includes only 250 single-file bugs. How do you plan to address the limited dataset scale and the exclusion of multi-file or cross-module bugs?”**
>
> We focus on single-file cases to ensure clean ground truth. For multi-file patches, not all edited files necessarily correspond to the true root cause, and labeling all of them as buggy would introduce ambiguity. In fact, such a setting is common in practice and aligns with widely-used SWE-bench Lite [1], where many agents are evaluated.
>
> Even under this constraint, agents face substantial difficulty on LinuxFLBench. We believe this work can serve as a necessary and reasonable starting point given the intricacy of the Linux kernel. Expanding the benchmark to include multi-file or cross-module cases is indeed valuable and is one of our key directions for future work. We plan to (i) release a multi-file track with carefully curated ground-truth file sets, and (ii) incrementally scale the benchmark over time by incorporating newly collected real-world bugs.
>
> **3: “Systematic analysis of agents' failures & Procedures for Ensuring No Data Contamination”**
>
> **Systematic Analysis of Failures in LLM Agents**
>
> Following the reviewer’s suggestion, we analyzed failure cases where all agents missed the buggy file within top-10 predictions.
>
> - *Directory-Level Accuracy on Failure Cases.* We checked whether agents at least predict the correct directory. Even in challenging cases, some agents can correctly infer the relevant directory. However, they tend to confuse files within the same directory.
>
>     | Agent         | Top-1 Dir Accuracy | Top-10 Dir Accuracy |
>     | ------------- | ------------------ | ------------------- |
>     | SWE-agent     | 0.21               | 0.29                |
>     | AutoCodeRover | 0.20               | 0.29                |
>     | Agentless     | 0.33               | 0.50                |
>
>
> - *Overlap Between Issue Descriptions and Location Information.* Following [2], we compared location-cue overlap between the failure cases and other cases, distinguishing (i) *straightforward* reports that explicitly mention the buggy file path and (ii) *challenging* reports that contain no related location keywords. The results show that failure cases are more likely to lack location cues, indicating a larger semantic gap that expands the search space and makes localization harder.
>
>     | Cases         | No Related Keywords | Exact Location Mentioned |
>     |--------------|---------------------|--------------------------|
>     | Failure Cases| 82%                 | 0%                       |
>     | Other Cases  | 48%                 | 1%                       |
>
>
> **Procedures for Ensuring No Data Contamination**
>
> We acknowledge contamination is inherently difficult to avoid for any benchmark derived from public open-source data. Importantly, even if LLMs have been exposed to Linux-related corpora, their performance on LinuxFLBench remains substantially lower than on general software benchmarks, which further highlights the difficulty and practical value of evaluating FL on complex systems like the kernel.
>
> That said, we still adopt measures to minimize potential leakage.  Mail augmentation enforces strict temporal filtering (emails strictly before the bug report) and removes emails with explicit Bugzilla references, preventing trivial exposure of ground-truth locations.
>
> ---
> References:
>
> [1] SWE-Bench: Can Language Models Resolve Real-World Github Issue? Jimenez et al. ICLR 2024.
>
> [2] Xia C S, Deng Y, Dunn S, et al. Agentless: Demystifying llm-based software engineering agents[J]. arXiv preprint arXiv:2407.01489, 2024.

---

### Note · Authors · 2026-01-05

I have read and agree with the venue's withdrawal policy on behalf of myself and my co-authors.